# Integrative CRISPR Activation and Small Molecule Inhibitor Screening for lncRNA Mediating BRAF Inhibitor Resistance in Melanoma

**DOI:** 10.3390/biomedicines11072054

**Published:** 2023-07-21

**Authors:** Sama Shamloo, Andreas Kloetgen, Stavroula Petroulia, Kathryn Hockemeyer, Sonja Sievers, Aristotelis Tsirigos, Ioannis Aifantis, Jochen Imig

**Affiliations:** 1Chemical Genomics Centre of the Max Planck Society, 44227 Dortmund, Germany; sama.shamloo@mpi-dortmund.mpg.de (S.S.); stavroula.petroulia@mpi-dortmund.mpg.de (S.P.); 2Max Planck Institute of Molecular Physiology, 44227 Dortmund, Germany; sonja.sievers@mpi-dortmund.mpg.de; 3Department of Pathology, Laura and Isaac Perlmutter Cancer Center, New York University School of Medicine, New York, NY 10016, USAkathryn.hockemeyer@nyulangone.org (K.H.); aristotelis.tsirigos@nyulangone.org (A.T.); ioannis.aifantis@nyulangone.org (I.A.); 4Laura and Isaac Perlmutter Cancer Center, New York University School of Medicine, New York, NY 10016, USA; 5Compound Management and Screening Center, 44227 Dortmund, Germany; 6Applied Bioinformatics Laboratories, Office of Science and Research, New York University School of Medicine, New York, NY 10016, USA

**Keywords:** lncRNA, melanoma, drug resistance, CRISPR activation, small molecule inhibitor

## Abstract

The incidence of melanoma, being one of the most commonly occurring cancers, has been rising since the past decade. Patients at advanced stages of the disease have very poor prognoses, as opposed to at the earlier stages. The conventional targeted therapy is well defined and effective for advanced-stage melanomas for patients not responding to the standard-of-care immunotherapy. However, targeted therapies do not prove to be as effective as patients inevitably develop V-Raf Murine Sarcoma Viral Oncogene Homolog B (BRAF)-inhibitor resistance to the respective drugs. Factors which are driving melanoma drug resistance mainly involve mutations in the mitogen-activated protein kinase (*MAPK*) pathway, e.g., *BRAF* splice variants, neuroblastoma RAS viral oncogene homolog (*NRAS*) amplification or parallel survival pathways. However, those mechanisms do not explain all cases of occurring resistances. Therefore, other factors accounting for BRAFi resistance must be better understood. Among them there are long non-coding RNAs (lncRNAs), but these remain functionally poorly understood. Here, we conduct a comprehensive, unbiased, and integrative study of lncRNA expression, coupled with a Clustered Regularly Interspaced Short Palindromic Repeats/Cas9-mediated activation (CRISPRa) and small molecule inhibitor screening for BRAF inhibitor resistance to expand the knowledge of potentially druggable lncRNAs, their function, and pave the way for eventual combinatorial treatment approaches targeting diverse pathways in melanoma.

## 1. Introduction

Despite comprising less than 1% of all skin-originated cancers, malignant melanoma exhibits the highest mortality rate among all skin cancer cases, exceeding 80% [1,2]. Both gender and age have an effect on the incidence of melanoma. In recent years, there has been an observed dramatic increase in the incidence of melanoma among the population aged 60 years and older. Conversely, age group between 15 and 24 showed a 5% reduction in the number of observed incidences. Currently, the median age at diagnosis of melanoma is 56 for women and 61 for men [3,4]. The primary risk factor for melanoma is ultraviolet radiation (UVR), which induces genetic mutations in melanocytes [5]. In particular, over 50% of patients harbor a gain-of-function mutation in proto-oncogene *BRAF* [6,7], leading to its hyperactivation that subsequently dysregulates the mitogen-activated protein kinase/extracellular signal-regulated kinase (MAPK/ERK) signaling pathway. This dysregulation increases cellular proliferation and survival while decreasing cell apoptosis [8]. In approximately 50% of *BRAF*-mutated cases, codon V600 of the *BRAF* gene (*BRAF^V600E^*) is affected [9]. Current approaches, such as targeted therapy using FDA-approved BRAF inhibitors (vemurafenib, dabrafenib, and encorafenib), either alone or in combination with MAPK/ERK kinase inhibitors or immune checkpoint inhibitors (ICIs) [10], as well as other immunotherapy approaches, have improved overall survival in patients with *BRAF*-mutated melanoma [1,9]. However, despite these advancements, mortality rates remain high because most patients eventually develop resistance to targeted therapies such as BRAF-inhibitor (BRAFi) resistance, leading to tumor progression. Though immunotherapies show higher response rates, a subset of patients do not respond and a fraction of the responders must interrupt treatment because of toxicity. The challenges in treating melanoma arise from its high biological complexity and increased mutational burden as the disease progresses, leading to elevated intra-tumor heterogeneity [9]. To better understand this complexity and develop more effective therapeutic strategies, it is crucial to uncover all underlying molecular mechanisms of drug resistance, including those that are poorly understood.

Among these underexplored areas, the study of long non-coding RNAs (lncRNAs) holds significant potential, as most studies on drug resistance center around protein-coding genes [11,12]. LncRNAs constitute a heterogenous group of molecules that lack protein-coding potential. They are involved in a wide variety of gene regulatory activities such as modifying chromatin conformation, altering nuclear organization, regulating transcription, and controlling post-transcriptional processes [13,14]. These multi-layered gene regulatory functions make lncRNAs indispensable participants in nearly all cellular processes, ranging from cell differentiation and proliferation to apoptosis. Moreover, their involvement extends to numerous pathological conditions, including cancer. It has been shown that lncRNAs can act as both tumor suppressors and oncogenes as well as influencers of drug resistance and sensitivity [12,14,15]. In melanoma, a few lncRNAs have been identified such as *SAMMSON*, *RMEL3*, and *LLME23*, which are expressed in more than 90% of melanoma subtypes. Their overexpression increases clonogenic potential primarily by acting on the MAPK pathway [16,17,18]. Recently, specific lncRNAs such as *SNHG5*, *MALAT1*, *HOTAIR*, and *SLNCR1* have been reported to promote metastasis and drug resistance in various cancers [19,20,21,22,23,24]. However, the mechanism or pathway through which these lncRNAs enhance the aggressiveness of cancer is still largely unknown.

The study of biologically relevant lncRNAs can be challenging due to their low expression level, complex function, and lack of protein expression. However, advancements in high-throughput techniques and bioinformatic methods, including CRISPR screening, have enhanced our ability to characterize lncRNA functions [15]. In this study, we integrated transcriptome analysis, Chromatin Immunoprecipitation Sequencing (ChIP-Seq), and CRISPR activation (CRISPRa-SAM) technology to systematically discover known and novel functional lncRNAs involved in BRAFi resistance in melanoma. Similar studies have been conducted; however, lncRNA are strictly cell type dependent and functionally promiscuous [25,26].

Therefore, our main aim herein is the urgent need to discover and complement lncRNAs in BRAFi and gain closer knowledge on resistance mechanisms but at the same time to provide a tailored strategy to streamline the experimental screening procedure with an integrative ‘Omics’, CRISPR, and small molecule inhibitor-based approach. Our working hypothesis is that lncRNAs overexpressed in BRAFi-resistant cell lines contribute to a gain-of-function phenotype; therefore, a focused screen would result in an increased hit rate. Applying these platforms, we were able to find various differentially expressed lncRNAs including novel ones, validated CRISPRa-identified lncRNA candidates in BRAFi resistance in melanoma, and portray their responsiveness profiles towards alternative pathway inhibition.

## 2. Materials and Methods

### 2.1. Cell Lines and Cell Culture

SKMEL-239 and SKMEL-239 T22/T23 BRAFi [PLX4032/vemurafenib]-resistant cell lines were provided by Eva Hernando Lab, NYU. Cells were cultured in RPMI medium (Thermo Fisher, Waltham, MA, USA) supplemented with 10% heat-inactivated fetal bovine serum (FBS, Thermo Fisher) and 1% penicillin/streptomycin (Thermo Fisher), in the presence of 5% CO_2_, and passaged regularly 1:3 while reaching 90% confluency. SKMEL-239 T22/T23 cells were maintained in the presence of 2 μM vemurafenib (PLX4032, Selleck Chemicals, Houston, TX, USA) during culture. HEK293T cells (ATCC #CRL-1573) were used for lentivirus production and were maintained in high-glucose DMEM (Thermo Fisher) supplemented with 10% heat-inactivated FBS and 1% penicillin/streptomycin and passaged every two to three days based on the confluency. Regular testing for mycoplasma contamination was performed on the cell lines using LookOut^®^ Mycoplasma PCR Detection Kit (Merck KGaA, Darmstadt, Germany).

### 2.2. Targeted Arrayed sgRNA Design, Synthesis, and Plasmid Library Preparation

Selected lncRNA genomic coordinates were extracted from in-house RNA-Seq data, which were then overlapped with histone marks (see RNA-Seq computational analysis). These sequences were imported into GuideScan, an online gRNA design and analysis tool [27], and the sgRNA target window was defined as described [28], between −250 and +50 bp of the respective (putative) TSS for optimal transactivation probability. Up to 10 sgRNA output was requested from the tool per lncRNA, in case the output was lower than 10, we filled the missing sgRNAs using the integrated CRISPR guide RNA design tool within Benchling (http://benchling.com) and selected the best sgRNAs after ranking them based on on-target and off-target scores. The library design included also 93 positive sgRNAs [26,29] targeting lncRNAs and protein-coding genes and 50 negative sgRNA controls, as listed in Appendix A. The targeted arrayed sgRNA oligo pool was synthesized by Twist Bioscience and amplified using array primers (Appendix A). The amplified oligos were then cloned in bulk into pLenti sgRNA MS2-puro-optimized backbone (Addgene #73797), a lentiviral sgRNA GFP-tagged vector, using Gibson Assembly Master Mix (New England Bioscience). The transformation process involved introducing 2 μL of the Gibson reaction into 50 μL of MegaX DH10B T1R Electrocomp™ Cells (Thermo Fisher), applying a voltage of 2 kV, resistance of 200 Ω, and a capacitance of 25 μF. We counted the colonies to evaluate the transformation efficiency and we estimated a yield of 100× coverage upon transformation. Subsequently, the sgRNA plasmid library was extracted from the bacteria using the QIAGEN Plasmid Maxi Kit (Hilden, Germany). To verify the distribution range, we performed NGS, ensuring that 90% of the sgRNAs fell within a 10-fold distribution range.

### 2.3. Lentivirus Generation

Recombinant lentivirus particles were generated by transient transfection of HEK293T cells with plasmids, as previously described [30]. Approximately 15 million HEK293T cells were seeded in a 15 cm plate one day prior to transfection to achieve 80–90% confluency the following day. For individual virus production, one 15 cm plate was used, while 10 × 15 cm plates were seeded for library-scale virus production. Each 15 cm plate was transfected with 22.5 µg of lentiCRISPR plasmid (vector of interest), 11 µg of VSVG (Addgene #8454), and 16.5 µg of pPAX2 (Addgene #12260) using 75 µg of Linear Polyethylenimine (PEI, Polysciences, Warrington, PA, USA). The virus supernatant was collected at 48- and 72-h post-transfection, filtered with a 0.2 µm Pore PTFE Membrane (Corning), and concentrated using centrifugal filter units (100 kDa MWCO, Merck KGaA). The concentrated viruses were immediately aliquoted and stored at −80 °C.

### 2.4. SAM Cas9 Expressing SKMEL-239 Stable Cell Line Generation

The SAM-dCas9 SKMEL-239 cell line was generated following the previously described method [29]. In brief, SKMEL-239 cells were transduced sequentially with lenti MS2-P65-HSF1_Hygro (Addgene #61426) and lenti dCAS9-VP64_Blast (Addgene #61425) vectors and were selected by treating them with Hygromycin (300 µg/mL) and Blasticidine (2 µg/mL) for 5 days, respectively. The expression of the dCas9 protein was evaluated using Western blotting with a dCas9 antibody (Cell Signaling), and HSF1 expression was quantified using qPCR (Appendix A). For all transductions, cells were seeded at 30% confluency in a 6-well plate. The following day, the culture medium was replaced with fresh media supplemented with 8 µg/mL of polybrene (Merck KGaA) and concentrated lentiviral particles. After approximately 16 h of transduction, the media was changed to remove excess virus particles and polybrene, and the selection process was initiated by adding the appropriate antibiotic.

### 2.5. RNA Extraction, Library Preparation, and Sequencing

Total RNA was isolated from 2 to 5 × 10^6^ cultured SKMEL-293 and SKMEL-239- T22/T23 find [PLX4032/vemurafenib]-resistant cell lines using the RNeasy Kit (Qiagen). The extraction process included on-column DNase digestion according to the manufacturer’s instructions. To prepare the library, rRNA was eliminated using the FastSelect-rRNA HMR Kit (QIAseq) following strand-specific cDNA library preparation with the Stranded Total RNA Lib Kit (QIAseq), as per the manufacturer’s protocol. The quality and quantity of the resulting cDNA library were assessed using Tapestation and sequencing was performed using the Illumina NextSeq 500 sequencer in the single-end method. Three biological replicates were processed for each sample, as described above.

### 2.6. RNA-Seq Computational Analysis

We identified differentially expressed novel and annotated lncRNA transcripts by analyzing the raw RNA-Seq fastq files using the lncRNA-screen pipeline (https://github.com/NYU-BFX/lncRNA-screen, accessed on 20 July 2017). This automated pipeline first evaluates the quality of the previously de-multiplexed fastq files and aligns them to the reference genome using STAR [31]. Subsequently, it performs transcript assembly, filtration, and classification to distinguish between annotated and novel transcripts. The platform also estimates the expression level and coding potential of the novel transcripts. To enhance the accuracy of identifying novel lncRNAs, we integrated ChIP-Seq data into the platform, utilizing its feature. Furthermore, by incorporating the DESeq2 r-package [PMID: 25516281], the pipeline provides reports on the differential gene expression of these novel transcripts among the samples and defines putative TSS, which is important for optimal sgRNA design.

### 2.7. Chromatin-Immunoprecipitation Sequencing (ChIP-Seq)

The experimental procedure in this study was based on previously described methods [32] with slight modifications. In summary, to prepare formaldehyde-fixed nuclei, 1% formaldehyde in PBS was added to 1 × 10^6^ cells and incubated at room temperature for 10 min. The fixation was quenched by adding Glycine to a final concentration of 0.125 M. The cells were then lysed using a cell lysis buffer (5 mM HEPES, pH 8.0, 85 mM KCl, 0.5% IGEPAL) supplemented with protease inhibitors (Roche) following vortexing, incubation on ice for 10 min, and spinning to pellet the nuclei. To generate the mononucleosomal particles, the pellet was resuspended in MNase digest buffer (10 mM NaCl, 10 mM Tris, pH 7.5, 3 mM MgCl_2_, and 1 mM CaCl_2_) with 1 U of micrococcal nuclease (USB), and incubated at 37 °C for 45 min. The reaction was stopped by adding EDTA to the final concentration of 20 mM, followed by a 10-min incubation on ice. The nucleus was spun down and resuspended in Nucleus lysis buffer (50 mM Tris-HCl, pH 8.1, 10 mM EDTA, pH 8.0, and 1% SDS) supplemented with proteinase inhibitors (Roche). Subsequently, the samples were sonicated using a Bioruptor (Diagenode, Denville, NJ, USA) at high intensity for 5 cycles (30 min on/30 min off) at 4 °C to obtain sheared chromatin fragments. Magnetic Protein G beads (Dynabeads, Thermo Fisher) were prepared by washing them with Citric-Phosphate Buffer (composition) and blocking them with 10 mg/mL BSA in citric phosphate buffer for 1 h. A total of 25 µL of beads per sample and 20 µL for pre-cleaning were prepared using this method. Approximately 200 µg of chromatin fragments were pre-cleaned with 20 µL of blocked beads and 10 volumes of IP dilution buffer (167 mM NaCl, 1.1% Triton X-100, 0.01% SDS, 1.2 mM EDTA, pH 8.0, and 16.7 mM Tris-HCl, pH 8.0) for 1 h at 4 °C. At this point, 1% of the input was reserved from the chromatin. Subsequently, 25 µL of blocked beads were coupled with 5 µg of Acetyl-Histone H3 (Lys27) (D5E4) XP^®^ Rabbit mAb or Tri-Methyl-Histone H3 (Lys4) (C42D8) Rabbit mAb (Cell Signaling, Danvers, MA, USA) for 4 h at 4 °C in IP dilution buffer. The antibody-coupled beads were then added to the pre-cleaned chromatin in IP dilution buffer and incubated overnight at 4 °C. After immunoprecipitation, DNA elution was performed by on-bead Proteinase K digest (Thermo Fisher) and overnight incubation at 65 °C under high agitation. Eluted DNA was then precipitated using ethanol and glycogen. cDNA Libraries were generated as described using Kapa Hyper Prep Kit (Roche, Basel Switzerland and Illumina TruSeq system, San Diego, CA, USA) and sequenced with NextSeq500.

### 2.8. Cell Viability Assay

To assess the impact of a single dose of vemurafenib, SAM-dCas9 SKMEL-239 cells were seeded in 48-well plates at a density of 1 × 10^4^ cells per well in 200 µL of RPMI medium. The next day, cells were infected with lentivirus particles containing a single sgRNA for each lncRNA candidate (3 sgRNA/lncRNA). After one day, the cells were subjected to a 14-day treatment with 2 µM vemurafenib/PLX4032 (Selleck Chemicals), during which fresh media containing the drug was replaced every two days; DMSO was used as a control. In another experimental setup, SAM-dCas9 SKMEL-239 cells were plated a in 96-well plate format (500 cell/well/100 µL medium) and different concentration of vemurafenib (0.001 µM to 50 µM) or DMSO was added to the sgRNA expressing cells for 7 days with a regular medium exchange during the drug treatment. At the end of the drug treatment procedure, Cell Proliferation Reagent WST-1 (Roche) was added directly to each well (10 µL/100 µL medium or 20 µL/200 µL medium) and the absorbance at 440 nm was measured using a Microplate Spectrophotometer (Tecan. Männedorf, Switzerland) after 3 h of incubation at 37 °C.

### 2.9. RNA Isolation and qRT-PCR

Total RNA was extracted using TRIzol RNA Isolation Reagent (Thermo Fisher), following the manufacturer’s instructions. RNA was reverse-transcribed to cDNA with SuperScript IV reverse transcriptase (Thermo Fisher) and after dilution, it was used as a template in the quantitative real-time PCR using SYBR Green Master Mix (Roche) in CFX96 (Biorad, Hercules, CA, USA) instrument. The GAPDH gene was used as the endogenous control and the 2^−ΔΔCT^ method was used to analyze the expression levels [33]. The primer sequences used in PCR are listed in Appendix A.

### 2.10. Western Blot Analysis

Cells were lysed and protein was extracted from cultured cells using RIPA buffer (Thermo Fisher) and the protein concentration was quantified using BCA Protein Assay Kit (Thermo Fisher). SDS-PAGE system (Biorad) was next used to electrophorese approximately 20 µg of protein per sample. Separated proteins were transferred to solid-phase membrane supports using the BIO-RAD blotting system and the primary antibodies (anti-Cas9 from Cell Signaling and anti-GAPDH from ProteinTech, Rosemont, IL, USA) were used to detect protein expression levels.

### 2.11. Small Molecule Inhibitor Cell Survival Assay (COMAS)

The compound screen was conducted at Compound Management and Screening Center (COMAS) using 57 available drugs (Appendix A) and DMSO as control. First, 5 × 10^5^ SKMEL-239-SAM_pool cells were transduced with lentiviral particles in a 6-well plate with the sgEGFR-positive control, sgNegCtrl, sgENSG182165_5, sgENSG224397_1, sgENSG226527_6 and sgXLOC_067371_5 puromycin selected (2 µg/mL final concentration) for three days as described above. On assay day cells were counted and 800 cells were plated into 384-well plates (Greiner). Individual drugs were added automatically at a single dose of 2 or 10 µM final concentration (Appendix A) in technical triplicates with the constant addition of 2 µM vemurafenib. Automated cell survival read-out was performed with CellTiter-Glo (Promega, Madison, WI, USA) three days later according to the manufacturer’s recommendation. Luminescence signals for each gene were normalized to sgNegCtrl and DMSO as 100% and plotted as a heat map using GraphPad Prizm. The sgEGFR-positive control is represented separately due to strong survival differences compared to the lncRNAs.

### 2.12. Statistics and Reproducibility

All statistical analyses were performed using GraphPad Prism (version 8.1.1 for macOS; GraphPad Software, Boston, MA, USA) using Student’s *t*-test, two-tailed. All sequencing and CRISPRa screening statistical processing were performed using DESeq2 r-package, lncRNA-Screen, and z-score normalization.

## 3. Results

### 3.1. Integrated Analysis of RNA-Seq and CHIP-Seq for lncRNA Discovery in Vemurafenib-Resistant Melanoma Cell Lines

We investigated the expression of long non-coding RNAs (lncRNAs) in two BRAFi-resistant clones derived from SKMEL-293 (SKMEL-239-T22 and SKMEL-239-T23) to identify novel and annotated lncRNAs potentially associated with BRAF inhibitor resistance. These cell lines show resistance to the antiproliferative effects of vemurafenib, making them a melanoma cell model for studying BRAFi resistance. In that sense, bulk RNA sequencing (RNA-Seq) analysis shows expressed lncRNAs in these cellular backgrounds. RNA-Seq combined with chromatin immunoprecipitation sequencing (ChIP-Seq) of active histone marks such as Histone 3 Lysine 4 trimethylation (H3K4me3) and Histone Lysine 27 acetylation (H3K27ac) will reveal bona fide active transcriptional units and highlight low abundant or novel lncRNAs. In addition, elevated lncRNA gene expression levels in BRAFi-resistant vs. parental cells accompanied by activating histone modifications suggest a potential biological function for a lncRNA [34]. We performed RNA-Seq and ChIP-Seq on parental SKMEL-239, SKMEL-239-T22 and SKMEL-239-T23 and integrated the output of these methods using an automated in-house lncRNA-Screen pipeline [35]. This combined approach enabled us to detect 162 overexpressed lncRNAs in T22 and T23 vs. SKMEL-239 parental cells (log2 fc > 0.5 and *p*-value < 0.05, Appendix A). These lncRNA candidates additionally harbor either a H3K4me3 or H3K27ac histone activation mark at the proximity of their respective TSS present in the cell line with higher lncRNA expression, which helps with the identification of a bona fide (novel) transcriptional unit. This filtering step involved intersecting the data from T22 and T23, resulting in only 12 commonly overexpressed lncRNAs (Figure 1a,b). This finding suggests potential differences in the drug-resistant mechanisms between these two cell lines, directly affecting the lncRNA expression pattern. We therefore collectively considered all 162 overexpressed lncRNAs in T22 and T23 as potential candidates for a CRISPRa screening in the SKMEL-239 parental cell line, aiming to assess their effect on melanoma BRAFi resistance phenotype.

Finally, our in-house pipeline LncRNA-screen has proven effective in identifying expressed novel and annotated lncRNAs within this particular pathological context. As an example, an IGV track of RNA-Seq and both H3K4me3 and H3K27ac ChIP-Seq of a newly identified lncRNA XLOC_002694 (genomic coordinates chr1:117236760-117253132), which is overexpressed in both melanoma BRAFi-resistant cells is shown in Figure 1c.

### 3.2. Targeted CRISPRa Screening Approach to Identify Putative lncRNAs Involved in Vemurafenib Resistance

To systematically identify functional lncRNAs involved in vemurafenib resistance, we established a dCas9 synergistic activation mediator (SAM)-based CRISPR activation (CRISPRa) system [29] in SKMEL-239 parental cells. Next, we designed a single guide RNA (sgRNA) library targeting 162 lncRNAs overexpressed in T22/23 BRAFi-resistant cell lines intending to render parental cells towards a BRAFi-resistant phenotype (Figure 2a). For each lncRNA, 10 sgRNAs were designed targeting the TSS, resulting in 1840 sgRNAs. Additionally, we included positive control sgRNAs targeting known protein genes and lncRNAs that enhance the resistance of melanoma cells to BRAF inhibitors, including 16 lncRNA hits from Joung et al. [26]. The library contained 100 negative control sgRNAs as well (Appendix A). The functionality of the SAM activation system was demonstrated by qPCR expression analysis of dCas9 in two generated cell SKMEL-239-SAM cell lines (SKMEL-239-SAM-pool) and a single clone (SKMEL-239-SAM-F5). Additionally, the epidermal growth factor receptor (EGFR) and G protein-coupled receptor 35 (GPR35)-positive control protein-coding genes were analyzed alongside with a randomly selected lncRNA ENSG0000272168 in the cell lines of interest using various primer sets (Appendix A). *EGFR* and *GPR35* have been both previously demonstrated to have implications in BRAFi resistance, thus making them suitable positive controls for our study. We were able to achieve gene overexpression over 4 orders of magnitude for all Cas9, lncRNAs, and protein-coding genes (Appendix A). To assure the plasmid sgRNA library integrity and prevent random loss of sgRNAs during the cloning procedure, we conducted deep sequencing showing no gross fluctuations of the sgRNA library distribution (Appendix A).

To conduct the screen, we applied two biological replicates: a pool of engineered SKMEL-239 cells expressing dCas9-VP64 and P65-HSF1, along with a single clone (F5) derived from the pool (Figure 2c). These cells were selected with antibiotics for a duration of seven days. Subsequently, we introduced the sgRNA lentiviral library into the pooled cells (SKMEL-239-SAM-pool) and F5 single clone (SKMEL-239-SAM-F5) at the multiplicity of infection (MOI) of 0.3. The cells were then treated with puromycin for four days to select transduced cells. The same day, we initiated the screening process by exposing the CRISPRa cells to either 2 µM vemurafenib (PLX4032) or an equal volume of DMSO as a control sample. We harvested the cells at two time points: 7 and 14 days post-drug treatment, and evaluated global changes in sgRNA representation of lncRNAs before and after vemurafenib treatment using next-generation sequencing (NGS) (Figure 2a, screening outline).

We next performed Principal component analysis (PCA) on normalized log_2_ count expression of sgRNAs in treated vs. control at each time point and observed the most variation in sgRNA representation at day 14 in both pool and single clone samples (Figure 2b).

SgRNA distributions were measured at day 0 and DMSO control at day 7, as well as DMSO control at vs. vemurafenib treatment, both at day 7 to serve as further control comparisons. These comparisons revealed two key findings: (a) robust screening settings and no major random sgRNA variations, and (b) similar distributions of the time-matched DMSO control compared to screening, starting at day 0. Together with only slight sgRNA enrichments in both replicates at day 7 we decided to continue analyzing potential lncRNA hits involved in BRAFi-resistant cells at day 14 related to day 0. Hit selection parameters were defined by applying z-score normalization to the average CPM values of all ten sgRNAs. Since the cell pool replicate at both time points, day 7 and day 14, resulted in a lower hit rate (Figure 2c) bottom), we applied this data set as the “reference” for selection, aiming to increase the stringency of our analysis. We used a *p*-value cut-off of ≤ 0.05 in one of the two analyzed time points (Appendix A). Besides protein-coding or other positive control genes that ranked within the top ten, this method resulted in three top lncRNA candidate genes (two annotated and one novel lncRNA) derived from our CRISPRa screen. These three candidates were subsequently subjected to single sgRNA validation. In total we identified 15 lncRNAs matching the *p*-value cut-off in either of the four samples, which are potentially associated with BRAFi melanoma resistance. Six out of those 15 lncRNAs were previously annotated, while the remaining nine were novel lncRNAs. (Figure 1c and Appendix A). We additionally noted from our cpm Appendix A, that sgRNA_6 targeting ENSG00000226527 shows an exceptionally high enrichment score, greater than 6-fold enrichment in all replicates, placing it within the range of protein-coding-positive control genes. Thus, we decided to include this gene in the validation step. Unfortunately, our attempts to design other sgRNAs in this region to activate ENSG00000226527 were unsuccessful (not shown).

Based on this, we selected the following four enriched lncRNA loci for downstream analysis and validations: ENSG00000182165 aka *LINC00482* overexpressed in both T22/T23; ENSG00000224397 aka *PELATON* overexpressed only in T22; ENSG00000226527 overexpressed only in T23, and XLOC_067371 overexpressed in both T22/T23. The ten individual z-score normalized sgRNA distributions for the four candidate lncRNAs are depicted in Appendix A).

### 3.3. Single sgRNA Vemurafenib Resistance Cell Survival Validation

To validate the four selected lncRNA CRISPRa screen hits potentially involved in melanoma BRAFi resistance, we picked the top 2–4 sgRNAs from our screen for each candidate, except for sgENSG00000226527_6; for this one, only one sgRNA was used. Selected sgRNAs were individually cloned into our sgRNA expression backbone for further analysis. SKMEL-239_SAM_pool cells were transduced with respective sgRNA and negative control expressing lentiviruses and treated with vemurafenib at 2 µM concentration for 14 days. Two sgRNAs inducing for each *EGFR* and *GPR35* protein-coding genes known to overcome BRAF inhibition and permit drug resistance (Appendix A; [29,36]) served as positive controls. Normalized cell survival towards negative control after CRISPR activation was performed using colorimetric WST-2 assay in independent biological triplicates. The results showed strong lasting and significant cell survival for both positive controls compared to sgNegativeControl treatment ranging from approximately 150% to up to 324% (*p*-value at least < 0.05). All sgRNAs inducing the lncRNA candidates except one (sgXLOC_2) were capable to permit drug-dependent cell survival; however, to a smaller extent than our positive controls. The lncRNA-dependent cell survival ranged from approximately 120% (sgXLOC_1) and up to a maximum of 166% (sg224397_1) (*p*-values at least < 0.05 to 0.001, Figure 3a). This range of cell survival upon lncRNA CRISPRa lies well within reported other lncRNAs involved in melanoma BRAFi resistance [26]. To further confirm our findings, we performed a vemurafenib concentration-dependent cell survival assay in analogy to the single-dose assay determining the IC_50_ value in SKMEL239_SAM parental cells using each the top sgRNA (sgENSG182165_5, sgENSG224397_1, sgENSG226527_6, sgXLOC_067371_5) including the sgEGFR_1 as positive and negative controls from Figure 3a. This treatment rendered the cells more BRAF inhibition resistant compared to the negative control by approximately 2 fold (*n* = 2 IC_50_: negative control = 0.5198 µM [SEM = 0.0695 µM], sgEGFR = 1.146 µM [SEM = 0.146 µM], sgENSG182165_5 = 1.778 µM [SEM = 1.0274 µM], sgENSG224397_1 = 1.100 µM [SEM = 0.3836], sgENSG226527_6 = 0.8242 µM [SEM = 0.1313 µM], sgXLOC_067371_5 = 0.94 µM [SEM = 0.3012 µM]) supporting the foundation to identify lncRNAs involved in melanoma BRAFi resistance.

The corresponding RNA overexpression confirmation for both the EGFR/GPR35-positive and lncRNA candidates was conducted by qPCR analysis at both, CRISPRa screening or assay time points of day 7 (Appendix A) and day 14 (Figure 3c). Fold-change expression values were calculated versus sgNegativeControl and GAPDH housekeeping gene using all relevant sgRNAs as described in Figure 3a. For the novel lncRNA XLOC_067371 and ENSG00000182165, there were no unambiguous isoform transcripts described or determined by the LncRNA-screen tool, therefore, we used each two primer sets spanning two separate intron-exon borders. For the EGFR- and GPR35-positive controls, we achieved between several 100- and several 1000-fold significant induction rates in line with the previous set-up of the CRISPRa system at both time points (Appendix A). However, the overexpression of the lncRNAs was found to be quite diverse for time points, different sgRNAs, and primer sets. For ENSG00000182165 at day 7, however, even a modest but significant ~25% reduction or no expression change was seen, which was compensated at day 14 by a 25% (primer set 2, sgRNA_#1) and 58% increase by primer set 1, sgRNA_#1 combination. sgRNA#5 failed to induce this lncRNA explaining the lower BRAFi resistance phenotype in Figure 3a). XLOC_067371 gene upregulation was only significantly possible at day 7 by between 2.4 and 4.6 fold at a maximum for all sgRNA and primer combinations whilst at day 14 a significant drop was seen between 73% and 53%. However, this transient lncRNA repression was still sufficiently high for a sustained phenotype over time. ENSG00000224397 CRISPR activation was consistently successful for both time points, sgRNAs, and primer-set combinations. It must be noted that sgRNA#12 displayed the lowest overexpression consistent with the least resistant phenotype. The overexpression significantly ranged between 1.7 and 3.2 fold on day 7 and 2.1 and 6.5 fold on day 14. Lastly, there was strong significant upregulation of ENSG00000226527 lncRNA 7.6 fold at day 7 and still to a smaller extent at day 14 of ~1.2 fold.

These findings were further confirmed by measuring the normalized cell survival in the presence of vemurafenib after CRISPR activation in a different cell line 501mel using colorimetric WST-2 assay similar to Figure 3a in independent biological duplicates. In this context significant cell survival was reproduced for all lncRNAs including the EGFR-positive control; however, to a generally lesser extent with a significant survival range of cells after lncRNA induction only between 104% (sgXLOC_14) and 159% sgXLOC_1) (exception sg224397_4 with ~206% survival) possibly reflecting the very strong cell type-dependent function of lncRNAs [37,38]. We also observed quite some variation between the survival effect of the same sgRNAs in 501mel_SAM versus SKMEL-239_SAM cells (Figure 3d) in line with previous reports [26].

### 3.4. Broad Panel Inhibitor Screen to Identify Potential lncRNA Mode of Action and Rescue Therapeutic Windows

To grasp some insights into (a) potential combinatorial drug treatment approaches that may overcome the respective lncRNA-mediated BRAFi resistance and (b) potential biological mode of actions/functions, we conducted a large-panel small molecule pathway inhibition single-dose pilot screen at the Compound Management and Screening Center (COMAS) using 57 different clinically applied or experimental drug targeting diverse pathways (Appendix A) or the DMSO-negative control. SKMEL-239-SAM_pool cells were CRISPR-activated for each of the four lncRNA candidates, sgNegativeControl, and sgRNA_EGFR (used as a positive control) using the best guide RNAs for a period of two days. These cells were then seeded into a 384-well format and treated with either 2 or 10 µM concentrations of each drug listed in Appendix A, along with a constant supply of 2 µM vemurafenib, for three days. Cellular survival was monitored using a luciferase-dependent read-out and plotted as a heat map, normalized to sgNegativeControl and DMSO as 100% baseline survival (Figure 4).

We hypothesize that (1) if the cell survival rate is lower than that of the DMSO control, cells will be still sensitive towards that drug/pathway treatment, indicating a potential combinatorial targeting approach. (2) In the case of survival equal or close to 100% of DMSO level, cells might be insensitive towards that particular pathway inhibition (i.e., pathway either irrelevant for cellular context or pathway silenced). (3) With a survival rate greater than 100%, the enforced lncRNA overexpression is contributing to a gain-of-function effect, mediating a BRAFi resistance phenotype within the targeted pathway. Since we are targeting the MAPK pathway and others at various regulatory positions or protein members, we will be able to determine whether a specific lncRNA acts upstream or downstream of a particular target. Additionally, by spotting clusters of similar responsiveness among the lncRNA candidates, we are able to identify potential similarities in their modes of function. The latter read-out was clearly seen for induced lncRNAs XLOC_067371 and ENSG00000226527 with very similar cell survival patterns, particularly in response to pathway inhibitors such as Selisistat targeting the epigenetics pathway and AZD8055 targeting the PI3K/AKT/mTOR pathway. In the case of ENSG00000182165 knock-down, cells remained strongly sensitive to Erlotinib, an inhibitor targeting EGFR, and 5-Azacytidine. This indicates that these two pathways are unlikely to be mediators of the lncRNA’s effects in BRAFi resistance. ENSG00000224397 gene modulation did not lead to a very stratified cell survival or death pattern upon diverse pathway inhibition. A conclusion for its potential functionality or combinatorial treatment option, therefore, remains obscure. Interestingly, our assay readout showed that upregulation of XLOC_06737, ENSG00000226527, and ENSG00000224397 will keep cells sensitive for Trametinib targeting MEK1/2 and SCH 772,984 both targeting ERK downstream of BRAF, excluding them being involved in BRAF resistance on the MAPK pathway. In contrast, ENSG00000182165 upregulation rendered cells more resistant towards Trametinib treatment, suggesting a potential contribution of this lncRNA in MAPK pathway modulation.

In conclusion, our combined approach of lncRNA activation involved in melanoma BRAFi resistance, along with a small compound inhibitor screen, appears to be promising to better understand lncRNAs mode of action and may aid in the development of more efficient combinatorial treatment strategies.

## 4. Discussion

MAPK-targeted therapies, whether employed as monotherapy or combination therapy of BRAF and MEK inhibitors, have significantly increased the overall survival rate in metastatic melanoma cases; nevertheless, acquired resistance almost invariably emerges irrespective of the specific agents used or their combinations [39]. The mechanisms of acquired resistance are complex due to the heterogeneity of pathways involved in BRAFi resistance [40]. Therefore, conducting a more comprehensive study on the mechanisms of resistance specific to each applied kinase inhibitor becomes crucial. In this study, we specifically focused on vemurafenib as the first FDA-approved BRAF inhibitor used in treatment of advanced malignant melanoma. We successfully profiled differential lncRNA expression in vemurafenib -resistant melanoma cell lines. This profiling served as the basis for a targeted CRISPRa screen to functionally identify lncRNAs involved in resistance mechanisms. By this, we were able to newly identify 15 lncRNAs potentially contributing to BRAFi resistance. Among them, four were further validated by orthogonal assays. Additionally, we conducted a large-panel small molecule inhibitor pilot screen for selected lncRNA candidates. The purpose of this screen was to portray their sensitivity responses intending to gain 2-fold information: Firstly, to provide possibly complemental therapeutic options for BRAFi resistance in melanoma, and secondly, to gain further insights into the putative biological roles of candidate lncRNA genes as an initial procedure.

The first step of our work unveiled a quite heterogeneous differential lncRNA expression profile in two BRAFi-resistant melanoma cell lines compared to their parental origin. Since we have not fully characterized the mechanisms of BRAFi resistance of the in vitro generated cell lines, it seems plausible that two distinct mechanisms are contributing to their phenotype. Furthermore, this observation likely reflects the well-known fact that lncRNAs are typically highly specific to particular cell types or even cell lines [37,38,41,42]. It might even be a general feature that lncRNAs contributing to melanoma or other cancer drug treatment resistance resemble a broad continuum of expression and functional traits in different cellular or patient contexts [43]. Therefore, it is crucial to carefully consider a pre-screening of potential non-coding drug resistance players upfront.

Our main hypothesis was that overexpressed lncRNAs within the drug-resistant background serve as putative gain-of-function mediators. This led us to rationalize an approach at first to execute an expression-based pre-filtering step which would lead to a higher hit rate while at the same time lowering the experimental effort compared to whole lncRNA transcriptome CRISPR screens. Remarkably, this focused CRISPRa screen detected 15 out of 162 (=9.25%) lncRNAs attributed to the BRAFi resistance phenotype; while in a study by Joung et al. a larger panel CRISPRa screen of 10,504 lncRNAs resulted in the identification of 16 significant hits equal to 0.15% (RIGER tool FDR 0.05 [26]). The phenotypic rescue range observed for lncRNA CRISPR activation, quantified as relative cell survival in the presence of vemurafenib in BRAFi-resistant melanoma cells, ranged from approximately 150% to a maximum of ~200% (Figure 3). These values fell within the reported range by Joung et al. However, it is important to note that the cell survival upon drug treatment for lncRNAs was much lower (>200%) compared to protein-coding genes [29,44].

This raises the interesting question of how far lncRNAs play a substantial role in drug resistance in general and in melanoma BRAFi resistance in particular. Several available reports have demonstrated that certain lncRNAs can trigger chemoresistance in diverse cancers, primarily by dysregulating apoptosis and promoting DNA repair processes [45] For instance, downregulation of *ERIC* (E2F1-Regulated Inhibitor of Cell death) lncRNA in osteosarcoma cells increases the rate of apoptosis among the etoposide treated cells, indicating *ERIC* enrollment in etoposide resistance [46]. Similarly, *PDAM* (p53-dependent apoptosis modulator) and *CUDR* (cancer upregulated drug resistant) are two other examples of drug resistance-associated lncRNAs that inhibit apoptosis, promoting cisplatin resistance in oligodendroglial tumor and bladder cancer, respectively.

*DDSR1* (DNA damage-sensitive RNA1) lncRNA is also implicated in cisplatin drug resistance but it’s mechanism of action differs from other cisplatin resistance-mediating lncRNAs. It induces homologous recombination DNA repair, by interacting with BRCA1 and hnRNPUL1, in Non-small cell lung cancer. Additionally, studies have shown that *PCGEM1* (prostate cancer gene expression marker 1) inhibits apoptosis by suppressing Caspase 7 activation, leading to doxorubicin resistance in prostate cancer. A high expression of *HOTAIR* in breast cancer has also been directly linked to increased metastasis and drug resistance [47]. While cancer types show heterogenicity, it is reasonable to hypothesize that a wide variety of modes of action, mediated by lncRNAs, may be involved in BRAFi resistance. Indeed, up to now, with the exception of lncRNAs *SAMMSON*, *EMICERI* and *RMEL* [17,48,49] which act epigenetically on Hippo signaling, p32 and regulating the metabolism of mitochondria and positive regulation of PI3K and MAPK signaling, there is little strong evidence for the direct functional involvement of lncRNAs in melanoma BRAFi resistance [50].

Typically, lncRNAs and other non-coding RNAs are considered to be weaker gene regulators compared to protein-coding genes, eventually explaining the phenomenon mentioned earlier. We also cannot rule out the perception that lncRNAs might act in concert with each other or underlying synergistic effect mediating of their impact on drug resistance (Figure 1a) and as indicated by similar drug responsiveness profiles by ENSG00000226527 and XLOC67371 (Figure 4). The identification of a clear mechanism of action for our candidate lncRNAs is still pending. Whether our selected lncRNA candidates simply regulate cell growth, viability, fitness or apoptosis alone or if they also modulate other cell survival pathways in a cooperative manner, remains to be addressed. Further lncRNA network analyses efforts will contribute to answer these questions in addition to analyze, e.g., neighbor gene expression correlation as a starting point, which has been shown a common feature of lncRNA action [26,51].

Taking the previous orthogonal study of Joung et al. as a benchmark, we included all 16 significantly enriched lncRNAs hits as well as *SAMMSON* lncRNA [26,48,49] as positive controls. However, among our top screening hits, we could only observe TCONS_00026521 (Appendix A), highlighting once again the different functions of lncRNAs and their strong cell type specificity.

By knowing the mentioned challenges, however, our main objective was to expand the current knowledge of lncRNAs involved in BRAFi resistance and help potentially overcome their mediated phenotype. To achieve this, we coupled our approach with a small molecule inhibitor screen, which proved to be successful. This adaptable new strategy could serve as an exemplary approach for other resistance target screening, in particular, in different cancer entities and therapeutic modalities such as immunotherapy resistances.

Finally, the performed small molecule pilot screen revealed remaining vulnerabilities of BRAFi-resistant melanoma cells by lncRNAs and might be a useful approach, which could be applied more widely to address further treatment windows in BRAFi-resistant melanoma. However, individual pathway vulnerabilities would need to further verified and functionally characterized.

## 5. Conclusions

In conclusion, the present work opens additional valuable insights into non-coding RNA-mediated melanoma BRAFi resistance biology, could be an exemplary strategy to study cancer drug-resistant mechanisms and biology and could eventually disclose new avenues for therapeutic combinatorial interventions in the future.

## Figures and Tables

**Figure 1 biomedicines-11-02054-f001:**
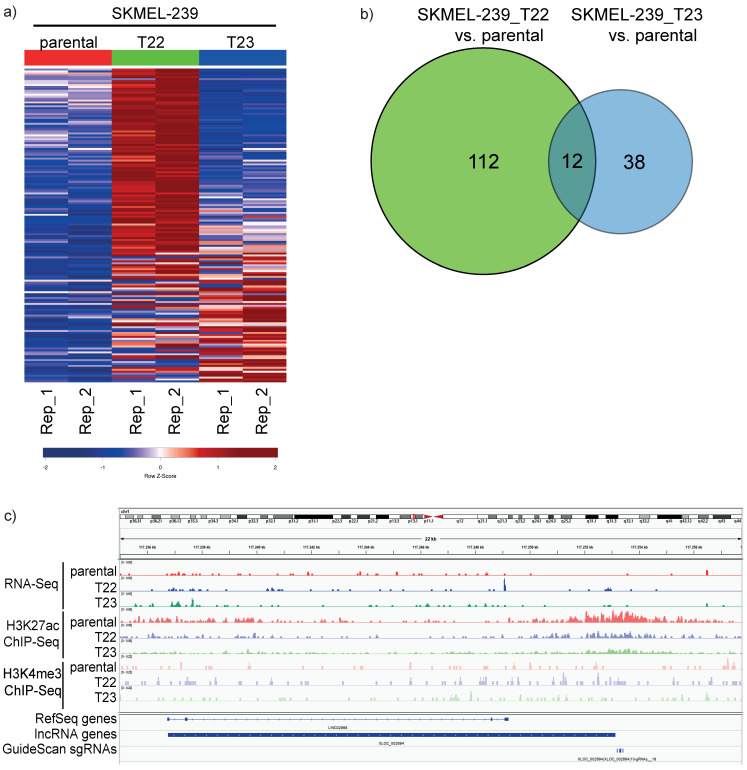
LncRNA expression profiling in melanoma and BRAFi-resistant cell lines. (**a**) Heat map comparison showing differentially expressed lncRNAs in SKMEL-239 parental, T22 and T23 vemurafenib-resistant cell lines represented as normalized z-score. Two replicates are shown in columns. (**b**) Venn diagram numbers of differentially expressed lncRNAs in SKMEL-239 parental versus T22 and T23 cell lines. (**c**) IGV genome browser tracks representing one replicate out of three for RNA sequencing, H3K27ac and H3K4me3 ChIP-Seq for a newly identified lncRNA expressed in SKMEL-239 parental, T22 and T23. RefSeq and new lncRNA as well as sgRNAs are shown as blue bars at the bottom.

**Figure 2 biomedicines-11-02054-f002:**
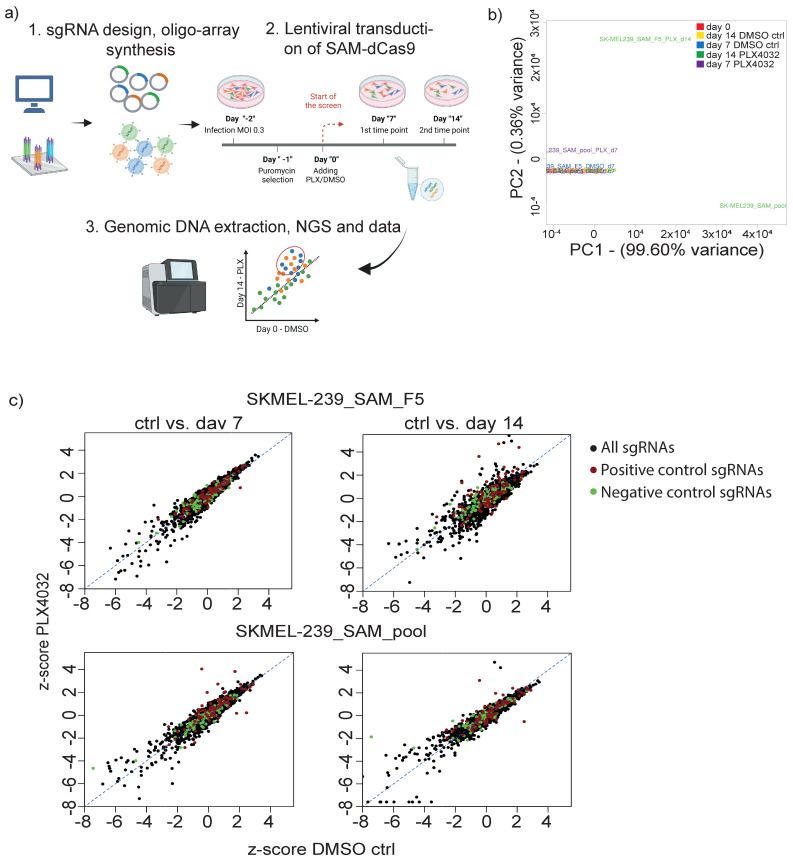
CRISPR activation screen for lncRNAs functionally involved in melanoma BRAFi resistance. (**a**) Sketch of screening outline (created with BioRender.com). (**b**) PCA analysis of sgRNA distribution for all samples and controls. (**c**) Scatter plots show the result of altered sgRNA library distribution in two biological replicates (SKMEL-239_SAM_pool, top and single clone F5, bottom) at time point day 7 (left) and day 14 (right) as shown as z-score normalized cpm sgRNA counts DMSO control versus PLX4032/vemurafenib treatment (constant dose 2 µM). green dots: sgRNA-negative controls, red dots: sgRNA-positive controls, black dots: all library sgRNAs.

**Figure 3 biomedicines-11-02054-f003:**
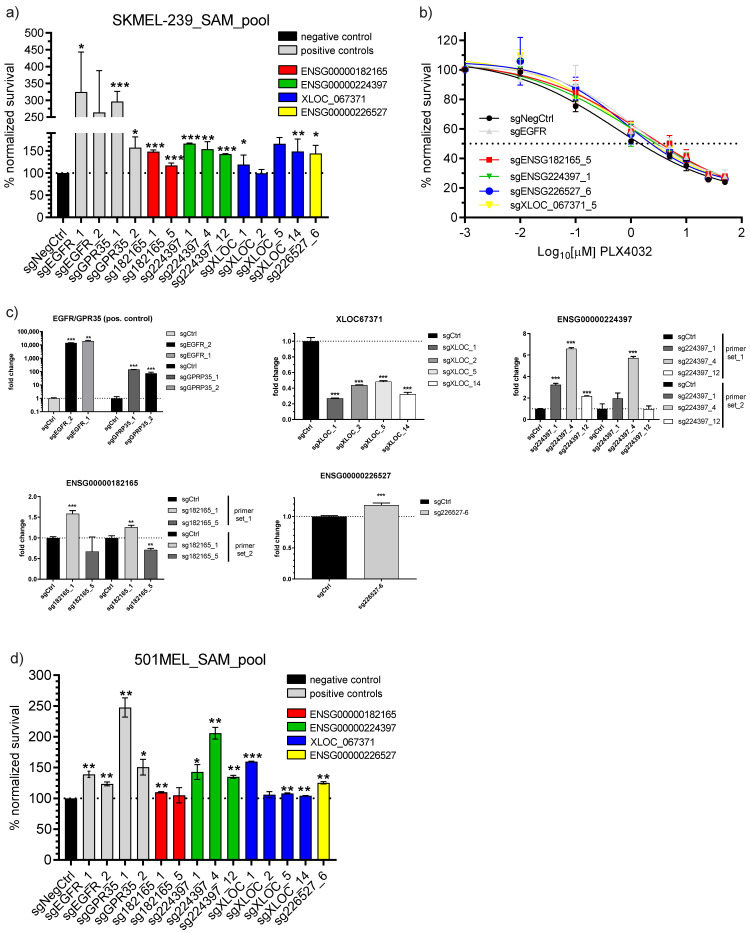
LncRNA-mediated BRAFi resistance validation. (**a**) WST-2 colorimetric cell survival assay showing percentage of surviving cells of selected sgRNAs activating lncRNA hits from CRISPRa screen compared to sgNegCtrl after 14 days of 2 µM PL4032/vemurafenib treatment in SKMEL-239_SAM_pool cells. *n* = 3 biological replicates. (**b**) Concentration-dependent killing curve of SKMEL-239_SAM_pool cells of best sgRNAs from (**a**) inducing lncRNA expression, sgNegCtrl (black) and the sgEGFR-positive control (grey) to determine the IC_50_ after seven day of drug treatment. *n* = 3 biological replicates. (**c**) qPCR expression validation of all lncRNA candidates and the EGFR-positive control after 14 days using diverse primer sets for some lncRNAs. (**d**) WST-2 colorimetric cell survival assay showing percentage of surviving cells of selected sgRNAs activated lncRNA hits from CRISPRa screen compared to sgNegCtrl after 14 days of 2 µM PL4032/vemurafenib treatment in 501mel_SAM_pool cells. *n* = 2 biological replicates. Student’s t-test two-sided: * *p*-value < 0.05, ** *p*-value < 0.01, *** *p*-value < 0.001.

**Figure 4 biomedicines-11-02054-f004:**
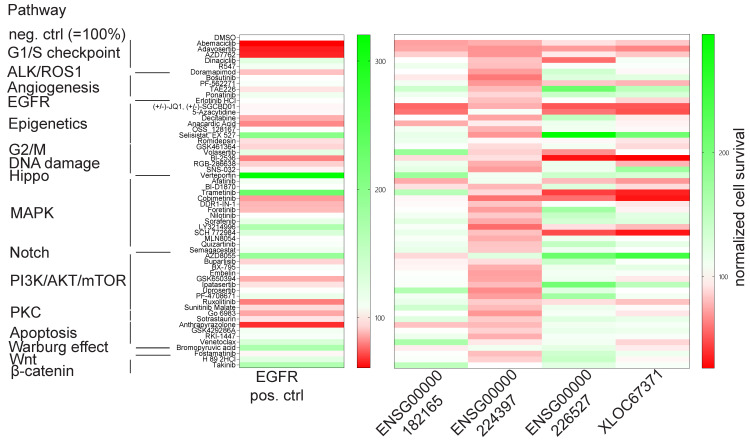
Prototypic small molecule inhibitor screen coupled with CRISPR-activated lncRNAs. Heat map showing cell survival of induced lncRNA candidates using best sgRNAs in response to various cellular pathway inhibitors normalized to sgNegCtrl and DMSO as 100%. EGFR induction serves as positive control.

## Data Availability

RNA-Seq and ChIP-Seq data will be made available upon request.

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
