# Peer review of "Integrative CRISPR Activation and Small Molecule Inhibitor Screening for lncRNA Mediating BRAF Inhibitor Resistance in Melanoma"

_biomedicines, 2023, doi:10.3390/biomedicines11072054_

Round 1
Reviewer 1 Report
In a study "Integrative CRISPR-activation and small molecule inhibitor
screening for lncRNA mediating BRAF inhibitor resistance in melanoma" the Authors aim at identifying lncRNA involved in drug resistance. Overall, the manuscript is well written, and the study presents novel data.
Two specific comments:
1. The manuscript requires a thorough editing. Figures are not sufficiently well organized, and they are not clear.
2. The Authors should justify more convicingly the idea of studying resistance to BRAF inhibitor alone, as combinations of BRAF and MEK inhibitors are used in clinics.
Reviewer 2 Report
In this manuscript, the authors conducted a comprehensive study on CRISPR-activation of lncRNA expression, coupled with small molecule inhibitor screening for BRAF inhibitor resistance in melanoma. They employed an integrative omics approach to identify differentially expressed lncRNAs, which were subsequently validated using the CRISPRa system. Additionally, the authors combined lncRNA activation with a small compound inhibitor screen to gain insights into the mode of action of the identified lncRNAs. Overall, this study is well-designed and provides clear results. However, there are a few concerns that should be addressed to further enhance the quality of this study.
Major concerns:
1. The authors stated that they detected overexpressed lncRNAs in T22 and T23 compared to the parental cells. However, in Figure 1c, the expression pattern (RNA-seq) appears to be inconsistent between T22 and T23, with T23 showing increased expression and T22 showing decreased expression. The authors should provide an explanation for this discrepancy or present more representative lncRNA data to clarify the findings.
2. For the cell survival experiments in Figure 4, it would be beneficial to include information about the number of replicates performed to ensure statistical robustness. Displaying the experimental replicates would enhance the reliability of the results.
Minor concerns:
1. In Figure 2b, it would be helpful to clearly label the samples in the PCA analysis for better interpretation.
2. In Figure. 2c, top candidates should be labeled in the scatter plot.
Reviewer 3 Report
Dear authors
This is an interesting and well-written manuscript in the field of skin cancer therapy. There are some comments to improve the work.
Abstract:
Please define “BRAF”, “NRAS” and “CRISPR” for first writing.
Introduction:
Please define “MAPK/ERK”, EGFR, GPR, …
Also, if various age groups are at different risk of melanoma, please state.
Methods:
Methods are appropriate. References are missing and need to be added.
Line 206: “104” cells writing should be corrected. Also line 224, 2-ΔΔCt. Please check the whole manuscript.
Results:
Ok
Discussion
Line 535: Please define s ERIC, DDSR1, PCGEM1, PDAM
and CUDR
this section can be improved by more details and comparison to other studies.
Kind regards
Dear editor
The English writing of the manuscript is appropriate.
kind regards
